# Impacts of the quinone-functionalized biochar on anaerobic digestion: Beyond the redox property of biochar

Qian Jiang[1,2]*, Wentao Zhou[1], Yue Chen[1], Zhenglong Peng[1], Chengcheng Li[1]

**1** School of Biological and Materials Engineering, Suqian University, Suqian, China, **2** Jiangsu Engineering Research Center of Novel Functional Film and Technology, Biological and Materials Engineering, Suqian University, Suqian, China

* jiangqian@squ.edu.cn

## Abstract

Recent developments in biochar materials have led to renewed interest in biochar modification for environmental applications, however, much uncertainty still exists about the impact of engineered biochar on a given biotechnological process. The redox properties of biochar were considered to be the key property for enhancing the methanogenic process, and the redox activity of biochar was closely related to the type and amount of oxygen-containing functional groups, especially quinone groups. Therefore, anthraquinone-2-sulfonate (AQS) was immobilized on algal biochar (ABC) by surface doping method, and the impacts of the quinone-functionalization process on algal biochar for regulating methane production were investigated in this study. Results showed that the immobilization capacity of AQS on ABC (ABC-AQS) reached 0.289 mmol/g. The acidogenesis rate was improved by 26.3% with the addition of ABC-AQS during anaerobic digestion test. However, methane production was inhibited rather than enhanced by the ABC-AQS, which could be attributed to the strong acid treatment stage involved in the biochar modification process. pH interferences, the generation and/or dissolution of inhibitory substances, and the release of $Zn^{2+}$ should be the major mechanisms of microbial inhibition by ABC-AQS. The findings of this study give us an important clue that when designing a biochar modification procedure for anaerobic digestion, attentions should be paid to the possible influences of chemical side reactions during biochar modification process on subsequent microbial metabolism, which would be valuable in designing engineered biochar for practical applications.

## Introduction

Anaerobic digestion (AD) is a mature technology for the treatment of organic waste, producing renewable energy in the form of methane and other biochemical products [1,2]. In recent years, there has been an increasing interest in improving the AD process with the addition of various biochar materials [3–5]. It is believed that a combination of waste reduction and energy recovery could be achieved through the biochar-enhanced anaerobic digestion process, which is of great significance for the circular economy, carbon neutrality, and other practical global issues [6,7].

**Data availability statement:** All relevant data are within the manuscript and its Supporting Information files.

**Funding:** This research is financially supported by the Suqian Sci&Tech Program (Grant NO. K202333) and the Scientific Research Foundation of Suqian University (NO.106-CK00042/060). The funders had no role in study design, data collection and analysis, decision to publish, or preparation of the manuscript.

**Competing interests:** The authors have declared that no competing interests exist.

Recent evidence suggests that the methanogenic improvements during anaerobic digestion were closely related to several properties of the biochar materials, the proposed biochar properties mainly included conductivity, porosity, and redox properties [8–10]. Among them, the redox properties were considered to be the key property for enhancing the methanogenic process by most of the researchers [10–12]. Moreover, the redox activity of biochar materials was influenced by the type and quantity of the oxygen-containing functional groups, especially the quinone groups [13,14]. The quinone group, as the most important redox group, endows biochar with the property of an electron shuttle, which can reversibly obtain electrons from microorganisms and transfer them to specific electron acceptors [15], thus enabling the cyclic transfer of electrons between different microflora during anaerobic digestion. Therefore, biochars containing abundant electroactive quinoid functional groups should be of higher biogeochemical and envirotechnical relevance.

Moreover, the application of biochar generally depends on its properties and costs, biochar was often modified to enhance target surface properties in practical applications [16–18]. It was believed that the surface functionality of the raw biochar that derived from the direct pyrolysis procedure was limited, such as poor porosity and limited redox activity [12]. However, raw biochar could be modified to improve its performance for applications, using a variety of physical and chemical techniques, including porosity activation, surface doping, and surface oxidation. Therefore, biochar was reported to be a platform carbon material for the design and synthesis of various functional materials [19], and biochar modification was considered to be an effective strategy for enhancing its environmental applications at a reduced cost. For instance, both hydroxyl and carboxyl groups on the biochar surface were all reported to be enhanced by torrefaction methods, thus helping to remove both organic and inorganic pollutants from water [20,21]. Moreover, Alvarez et al. [22] have reported a method for loading quinone functional groups (anthraquinone-2-sulfonic acid, AQS) onto the surface of granular activated carbon, then the reduction and the removal of Congo red in the wastewater were significantly promoted by this modified char material.

Therefore, it can be hypothesized that the redox activity of biochar materials could be enhanced by immobilizing electroactive quinoid functional groups on its surface, and the functionalized biochar should be able to serve as an electron conduit to enhance syntrophic relationships between different microbial groups during anaerobic methanogenesis. In addition, it should be noted that most studies in the field of biochar modification have focused on the removal of organic and inorganic pollutants from water, soil, and other systems [16–18,22,23], while the knowledge of engineered biochar on a given biotechnological process (e.g., anaerobic digestion) is limited. The impact of quinone-functionalized biochar on anaerobic digestion is worthy of further exploration.

Compared to surface oxidation methods, the target functional groups could be selectively immobilized on the surface of biochar by the surface doping method. Therefore, in this study, algal biochar (ABC) was firstly derived from the Taihu blue algal biomass which is a biomass solid waste from water treatment in Wuxi (China), and ABC was modified by the immobilization of anthraquinone-2-sulfonate (AQS) on the surface of ABC, then impacts of the quinone-functionalized biochar (ABC-AQS) on the anaerobic digestion of the model compound glucose were investigated, and mechanisms associated with the methanogenic performances with the addition of ABC-AQS were explored. The results of this study will provide a better understanding of the role and significance of the quinone-functionalization process on algal biochar for regulating methane production, which would be valuable for the design of engineered biochar for practical applications.

## Materials and methods

### Biochar and biochar modification

Due to harmful cyanobacteria blooms, Taihu blue algae was considered a kind of abundant and harmful biomass waste in the water treatment of Wuxi (China). In this study, blue algae biomass was collected from the algal blooming sites alongside Taihu Lake of Wuxi city, in which the dominated community was *Microcystis*. After filtration and cold drying, algae powder was obtained and stored at 4 °C before use; then algal biochar (ABC) was derived from the slow pyrolysis of the blue algae biomass in a lab-scale tube furnace (Shengli Instruments, China), the pyrolysis temperature was maintained at 450 °C for 2 h, the heating rate was 10 °C/min. Methods for the characterization of ABC was consistent with our previous studies [24,25]. More experimental details about the measurement of raw biochar samples, including pH, particle size, and specific surface area, were summarized in the section "Materials and methods" of Supplementary Information.

ABC was functionalized with anthraquinone sulfonate (AQS, Sigma-Aldrich) in this study (Fig 1a), and detailed procedures about the quinone-immobilization method were performed according to the previous reports [22]. In summary, firstly 25 g of ABC was exposed to a solution of 250 g/L of $ZnCl_2$ dissolved in the concentrated hydrochloric acid, the mixture was stirred and mixed for 24 h at room temperature (16–25 °C). Then ABC was separated from the mixture and washed with distilled water until the pH of the filtrate was neutral (pH 7.0). Secondly, the resultant biochar was immersed in a solution of AQS (pH 7.0, 5.3 mmol/L) for 24 h at room temperature (16–25 °C). The variation of AQS concentration during the modification process was measured with a spectrophotometer at 310 nm according to the standard curve method (Fig 1b and 1c). Then the modified biochar was separated again from the mixture by filtration, and the modified biochar was repeatedly rinsed and washed with the distilled water until the pH of the filtrate was neutral and a clear, transparent filtrate was obtained. After modification, the functionalized biochar was collected and dried in a drying oven at 60 °C overnight. Lastly, all the biochar samples were sealed and kept at room temperature (16–25 °C) before usage. And the functionalized biochar obtained by the quinone-immobilization method was named "ABC-AQS" in this study.

### Analysis of the raw and the functionalized biochar

The pH of different biochars was determined using the soil pH determination method. The cyclic voltammetry (CV) method was applied to test the redox characteristics of the biochar samples. A modified electrochemical method was used to quantify the electron exchange capacity (EEC) of the biochar samples, which includes the electron-accepting capacity (EAC) and the electron-donating capacity (EDC). The modified electrochemical method was consistent with our previous study [24]. The crystalline structure of biochar sample was examined by X-ray diffraction (XRD, D2 Phaser, Bruker, Germany) over a 2θ collection range of 10° - 80°. Other detailed information about the characterization of biochar samples, including the measurements of pH, specific surface area (SSA), iodine value and phenol adsorption value, CV and EEC quantitation, was described in the section "Materials and methods" of Supplementary Information.

### Setting for the anaerobic digestion tests

The impact of the raw and the quinone-functionalized biochar on the anaerobic digestion process was examined in batch operation. For the anaerobic digestion test, glucose was set to the sole carbon source for the anaerobic digestion test to avoid possible interference from substrates. The

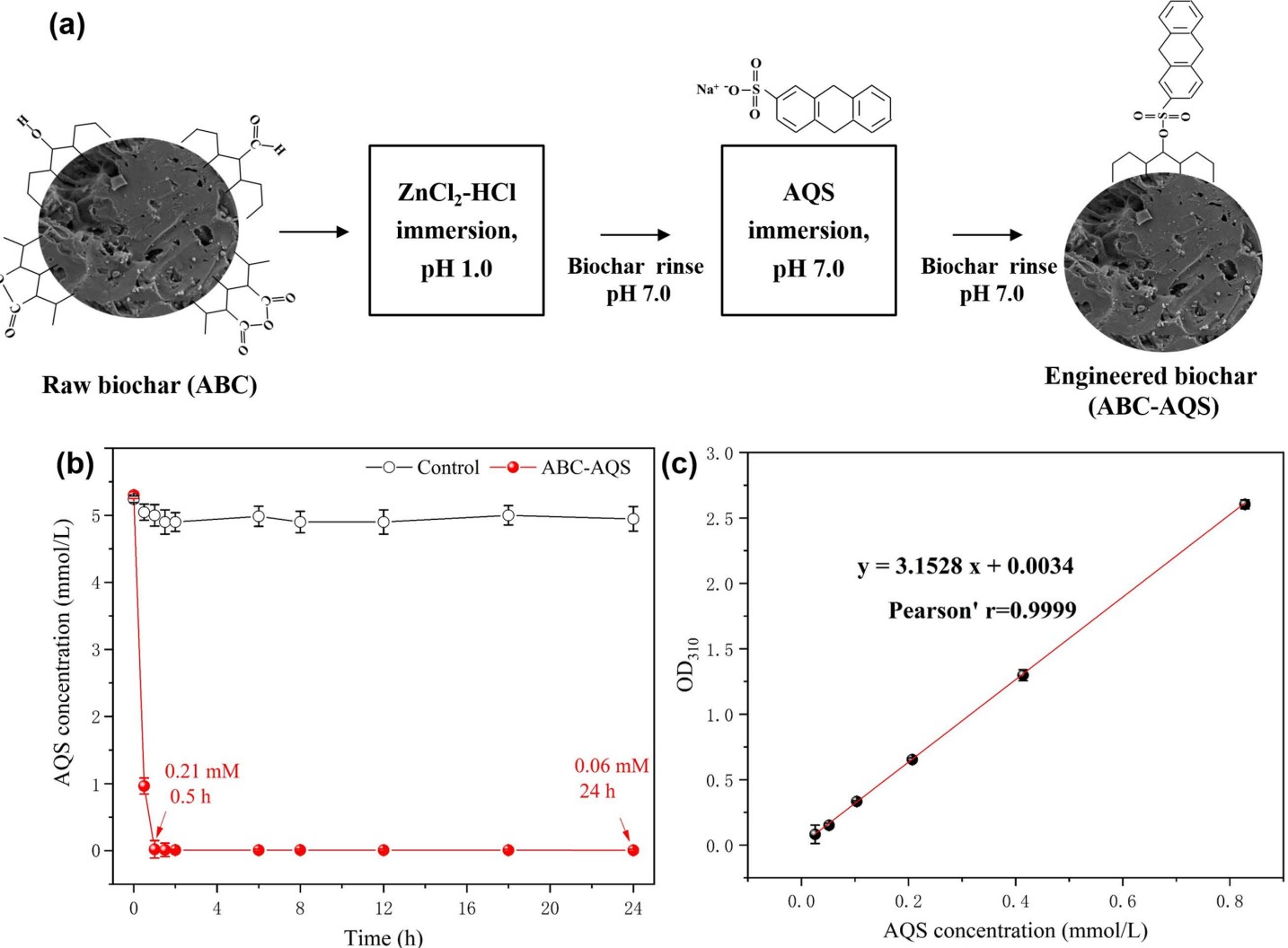

**Fig 1. The functionalization of algal biochar (ABC) with quinone groups (AQS). a, procedures and mechanisms; b, variation of AQS concentration during the modification process; c, the standard curve for quantitation of AQS during the modification process.**

batch anaerobic digestion tests were performed in 1.2 L custom glass reactor incubated at mesophilic temperature (37 ± 1 °C), the working volume was set to 600 mL. Each reactor contained 600 mL glucose solution (5 g/L), 6 ml of trace element solution, and 6 ml of vitamin solution. The composition of the trace element solution and vitamin solution was described in our previous study [24]. Sodium bicarbonate was added to each reactor to maintain the pH stability of the anaerobic digestion test and the concentration was set to 4 g/L. Ammonium chloride was applied to regulate the C/N ratio of the synthetic substrate to 25:1 in this study. Each reactor was inoculated with the anaerobic digested sludge collected from another anaerobic digestion reactor, and the initial inoculation ratio was set to 5% (v/v). The inoculum sludge reactor was operated continuously for more than 300 days, detailed information about the inoculum sludge and the reactor was explained in our previous study [26]. Before the anaerobic digestion test, the mixed substrate in each reactor was adjusted to 7.0 ± 0.5 by hydrochloric acid and sodium hydroxide solution.

In addition to the above basic substrates, the differences in biochar addition between the experimental and the control groups for anaerobic digestion tests were described as follows:

Compared to the control reactors without biochar addition, both the raw biochar and the quinone-functionalized biochar were added to the control and the experimental reactors, respectively. The biochar dosage was set to 10 g/L based on the working volume of the reactors. Hereto, the reactors amended with raw and functionalized biochar were named "ABC" and "ABC-AQS" groups respectively.

After initial mixing, all the reactors were internally aerated with high-purity nitrogen (99.99%) for 30 minutes to maintain an anaerobic environment. Then all reactors were sealed with matching plugs and incubated in a greenhouse, the temperature was set to 37 ± 1°C. During the anaerobic test, the mixture in each reactor was agitated continuously by a stirrer equipped with the custom reactor. Both gas and liquid samples were collected for the analysis of the methanogenic performance until the gas sampling was stopped on the 30th day. To ensure the reliability of the data, after the first batch of anaerobic digestion, the second batch of anaerobic digestion test was operated under the same conditions as the first batch operation, and the batch experiments in this study lasted for a total of two months.

### Model fitting analysis of anaerobic digestion process

A widely used model for analyzing methanogenic performance, the modified Gompertz model, was used to evaluate the impacts of the raw and the quinone-functionalized biochar on the anaerobic digestion process. Generally, three key parameters including maximum methane production potential ($P_m$, mLCH$_4$/g COD), maximum methane production rate ($R_m$, mL/g CH$_4$ COD d), and lag phase time ($\lambda$, d) were simulated for the comparison of the methanogenic performances with different biochars. More information including model formulas, resultant parameters, and citing references about this model could be found in the section "Materials and methods" of Supplementary Information, as well as in a series of other reports [4,26,27].

### Other analysis

For the liquid samples, firstly pH and chemical oxygen demand (COD) values were detected by the standard APHA method [27]; Moreover, to monitor the variation of the volatile organic acids in anaerobic reactors, the rest of the liquid samples were centrifuged, filtered and finally analyzed by the gas chromatography method (GC-2010, Shimadzu, Japan) [24]. For the gas samples, gas components were analyzed by the gas chromatography method (FULI 9790II, China) [25], then gas volumes were measured by displacement of the saturated aqueous sodium chloride in a graduated measuring cylinder, and all gas volumes were calibrated to the standard condition (273 K, 1 atm) for analysis in this study.

### Statistical analysis of data

The results were expressed as means and standard deviations. Figures were plotted with the Origin 9.5 software. Statistical analysis of data was performed by one-way ANOVA using SPSS 16.0 software. Data was processed and plotted by Excel and Origin software 9.5 respectively, and expressed as "mean ± standard deviation" in this study. Significant differences were tested using Duncan's multiple range test (P=0.05) and the correlation was analyzed with the Pearson test (two-tailed) at P=0.05. Any differences between the mean values at P<0.05 were considered statistically significant.

## Results and discussion

### Functionalization of algal biochar

In this study, ABC was pretreated by the ZnCl$_2$-HCl solution immersion to enhance the surface immobilization of AQS (Fig 1a). During the AQS immobilization process, the AQS

concentration of the biochar suspensions before and after modification was determined. It was found that the immobilization capacity of AQS on algal biochar reached 0.289 mmol/g biochar after 24 hours of modification (Fig 1b). The concentration of AQS was maintained stable in the control groups throughout the modification process, which confirmed that the reduced AQS was attributed to the surface immobilization of ABC (Fig 1b).

Moreover, a higher AQS immobilization capacity was obtained by activated carbon, achieving 0.79 mmol/g activated carbon in this study, which was 1.82 times higher than that of ABC. Alvarez et al. [22] also declared that the adsorption capacity of AQS could reach 0.98 mmol/g activated carbon. It was believed that the immobilization of AQS on biochar was mainly attributed to the anchorage with the sulfonic group ($-SO_3^-$) of AQS; Moreover, the adsorption capacity of AQS was highly relevant to the density of the hydroxyl groups (–OH) on the surface of biochar materials (Fig 1a). However, in addition to cationic exchange, physical adsorption originating from the porous structure should also be critical for the anchorage of AQS on biochar. During the modification process, ABC was separated and rinsed by distilled water until pH neutral after the $ZnCl_2$/HCl pretreatment and the AQS immersion. However, it was observed that the eluate still showed a brown color even after several rinses. Therefore, the higher adsorption capacity of AQS by activated carbon should be attributed to its more abundant porous structures.

## Effect of the functionalized algal biochar on methane production

The cumulative methane productions of anaerobic digestion tests with different biochars were shown in Fig 2. The anaerobic digestion tests were ended at day 30 since the daily biogas production was less than 10 mL. As shown in Fig 2a, the cumulative methane yields of the blank groups were 223.8 ± 5.1 mL/g $COD_{added}$, and the methane production of the ABC groups was increased to 239.5 ± 4.6 mL/g $COD_{added}$. Compared to the blank control, the cumulative methane yields were increased by 11.7% after the ABC addition. It has been widely accepted that the anaerobic digestion performance could be improved by the addition of biochar, and the methanogenic improvement by ABC in this study was also consistent with the results in our previous study [24,26]. On the contrary, methane production was completely inhibited by ABC-AQS since there was no methane production in the ABC-AQS groups throughout the anaerobic digestion test (Fig 2a and 2c).

Moreover, variations of the key parameters of the anaerobic processes have provided more evidence for the inhibitory effects of the engineered biochar. On one hand, the pH of the ABC-AQS group was observed to decrease rapidly from 7.0 at the beginning to nearly 5.5 in 3 days (Fig 2b). Methanogenesis was believed to proceed under the pH-neutral condition [28], whereas a rapid decrease in pH to acidic condition in the ABC-AQS groups was not favorable for the metabolism of methanogens.

On the other hand, as shown in Fig 2d, the VFA concentrations of the blank, the ABC, and the ABC-AQS groups were comparable at day 1 of the trials (1.8 ± 0.3 g COD/L), while the acidogenesis rate was improved 26.3% by ABC-AQS compared to ABC since the cumulative VFA concentration reached to 4.8 ± 0.4 g COD/L at day 5 (Fig 2d). It was believed that the presence of electron shuttles (e.g., quinones) was favorable to accelerate the redox reactions associated with organic degradation, which could explain the accelerated acidogenesis by ABC-AQS. Likewise, Yang et al. [29] also reported that the VFA production from proteins of sewage sludge was improved by anthraquinone-2,6-disulfonate (AQDS). Quinone groups originated from the modified biochar may have acted as electron shuttles during the acidogenesis stage of anaerobic digestion and thus favorable for the acid productions. However, the enhanced acidogenesis failed to enhance the subsequent methanogenesis in this study, the resultant VFAs were barely consumed in the ABC-AQS group compared to blank and ABC groups (Fig 2d).

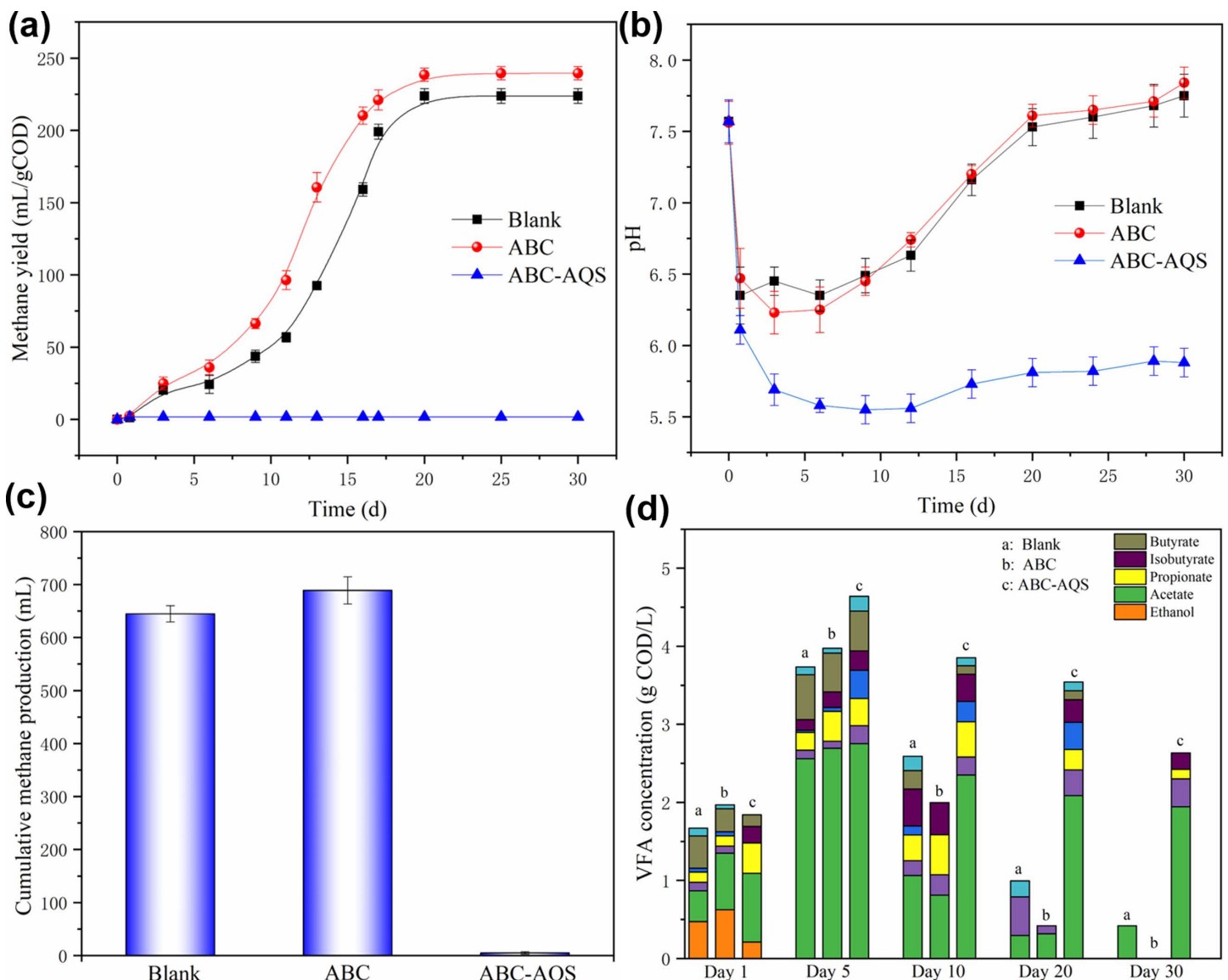

**Fig 2. The impacts of the functionalized biochar on anaerobic digestion process. a, methane yield; b, pH variation; c, cumulative methane production; d, variation of VFA concentration during anaerobic digestion.**

Overall, it can be concluded that the anaerobic digestion of glucose, especially the methanogenic process, was inhibited rather than enhanced by the quinone-functionalized biochar (ABC-AQS) in this study. The pH stability of the anaerobic digestion process was crashed due to the addition of ABC-AQS (Fig 2b). The acidogenesis was accelerated in the presence of the quinone groups on biochar (Fig 2d), which might have contributed to the rapid decrease in pH during anaerobic digestion with ABC-AQS (Fig 2b). Moreover, according to a previous study the pH interference that mediated by the acid-pretreated biochar have caused the disruptions of pH stability, and thus leading to the failure of anaerobic digestion [24]. Therefore, exploring the effect of the modification process on biochar properties would be beneficial for understanding the role and significance of the quinone-functionalization method on algal biochar for regulating the anaerobic digestion process.

### Redox activity of the functionalized biochar

The redox properties of biochar were analyzed and compared before and after the biochar modification process. As shown in Fig 3a, a couple of reversible redox peaks could be observed in the CV analyses of ABC samples, while both the oxidation and the reduction peaks were significantly strengthened after the functionalization with AQS (Fig 3b). It was clear that the redox properties of ABC were significantly altered due to the immobilization of quinone groups on its surface, indicating that the quinone-functionalized biochar could serve as both electron acceptors and donators during electrochemical reaction, and the microbial metabolism would benefit from the enhanced redox activity during anaerobic digestion [9,10].

Moreover, the electron exchange capacity (EEC) of different biochar samples were detected by the electrochemical method and results were shown in Fig 4. It was evident that the EAC of ABC-AQS samples were significantly decreased to 0.71 ± 0.34 μmol e⁻/g, compared to that of the raw biochar samples (1.20 ± 0.21 μmol e⁻/g) (Fig 4a). However, the EDC of ABC-AQS samples were 1.58 ± 0.28 μmol e⁻/g, which have increased by 113.5% compared to the raw biochar (0.74 ± 0.15 μmol e⁻/g) (Fig 4b). It is believed that the redox activity of biochar was closely related to its surface functional groups, and the improvement of EDC could be attributed to the immobilization of the quoined groups in this study. Jiang et al [24] also reported the electron-donating capacity of biochar was increased by 64.9% after surface modification, which could be explained by the increase of specific surface functional groups, e.g., phenolic and lactonic groups.

Compared to EAC, EDC is a key parameter for evaluating the redox activity of biochar, and a series of studies have reported that the methanogenic improvements were positively correlated with EDC of different biochar materials [24,30]. Therefore, the quinone functionalized biochar was supposed to be able to improve the methanogenic performance during anaerobic digestion. However, the methanogenic inhibition by ABC-AQS was observed in this study, indicating that the other factors beyond the EDC increase of biochar should have played a more dominant impact on the anaerobic digestion of glucose. Considering that the quinone groups are generally

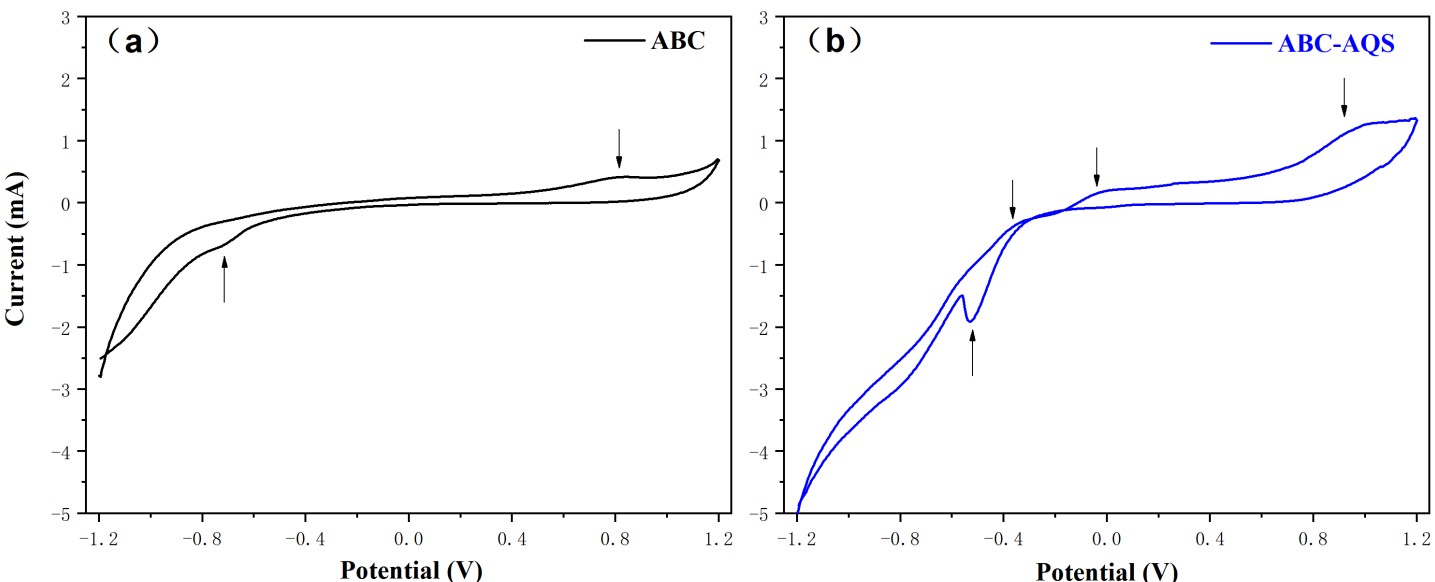

**Fig 3. The cyclic voltammetry analysis of the raw and the functionalized biochar. a, raw biochar; b, functionalized biochar.**

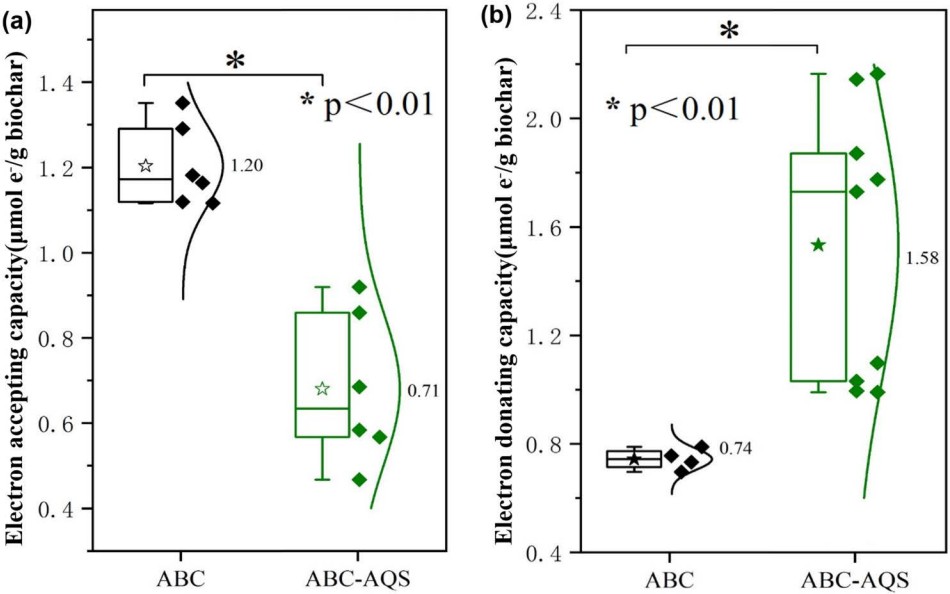

**Fig 4. The calculated electron accepting capacity (EAC) and electron donating capacity (EDC) of the raw (ABC) and the functionalized biochar (ABC-AQS). a, electron accepting capacity; b, electron donating capacity.**

considered important surface functional groups that are capable of both donating and accepting electrons during electrochemical reactions, the different variations in the EAC and EDC of ABC-AQS compared to ABC could give us an important clue for understanding the role of the quinone-functionalized biochar during anaerobic digestion. The possible reasons for the inhibition of anaerobic methanogenesis was discussed in the next section.

### Role of the quinone-functionalized biochar during anaerobic digestion

The redox properties of biochar are closely related to the type and quantity of its surface functional groups, and an increase of the biochar redox activity has been proven to be beneficial for the establishment of syntrophic relationships between different microbial species during the anaerobic digestion process [10]. Among various oxygen-containing groups, the quinone groups have been highlighted for their excellent redox activity. It was believed that biochars containing electroactive quinone functional groups would be favorable for anaerobic methanogenesis [13–15].

According to the results of electrochemical analysis in this study, the redox activity of ABC, especially the electron-donating capacity was significantly enhanced by the immobilization of AQS on its surface (Figs 3 and 4). However, variation in biochar properties was found to have an interesting effect on the subsequent anaerobic digestion process. On one hand, the acidogenesis rate during anaerobic digestion test was improved by 26.3% with the addition of the functionalized biochar, indicating that the enhanced redox activity of biochar was favorable for the microbial metabolism during anaerobic digestion; On the other hand, the methanogenesis stage of anaerobic digestion was inhibited by the modified biochar, since there was no methane production throughout the subsequent anaerobic tests (Fig 2). Moreover, further study confirmed that anaerobic methanogenesis was not significantly inhibited by the sole addition of AQS (2.8 mM), indicating that the inhibition of methanogenesis by ABC-AQS was mainly due to the modification process rather than the quinone groups that mentioned in this study.

The crystalline structure changes of the algal biochar before and after the quinone immobilization process were compared. As shown in Fig 5, the XRD spectra of the algal biochar all showed broad peaks at ~25.6°, indicating the amorphous crystalline structure of ABC. Moreover, the amorphous crystalline structure of the algal biochar was significantly enhanced after the quinone immobilization (Fig 5b). Previous studies have found that the amorphous crystalline structures of algal biochar were also enhanced after nitric acid immersion treatment, owing to the oxidation and pickling effects involved in the strong acid immersion treatment [24]. This finding also provided an important clue for analyzing the methanogenic inhibition by quinone-functionalized biochar in this study, because an important step in the modification process was to soak the algal biochar in concentrated hydrochloric acid for 24 h. The concentrated acid treatment process may have a significant effect on the biochar properties, which was confirmed by the changes in the crystalline structure of ABC after the biochar functionalization.

The brown-colored digestate was observed in the ABC-AQS group at the end of anaerobic digestion, which was distinct from the other reactors. It should be noted that there were still comparable amounts of organic matter remaining in the low-temperature pyrolysis biochar (450 °C), the residual polysaccharides, lipids, and proteins in the biochar may be reacted to generate inhibitory substances (e.g., heterocyclic aromatic hydrocarbons) during the acid immersion treatment. These inhibitory products may have an impact on the metabolic activity of microorganisms [31,32]. For example, Zhou et al. [33] also found that the treatment of corn-stover biochar with $H_2SO_4$ inhibited the methanogenic process of anaerobic digestion, Jiang et al. [24] have reported anaerobic methanogenesis were terminated and the acidogenesis process was prolonged in presence of $HNO_3$-modified biochar. The generation and/or dissolution of inhibitory compounds like benzene, phenols, and polycyclic aromatic hydrocarbons were reported to be enhanced due to the acidic treatment process, which could have explained the negative effects of ABC-AQS in this study.

Moreover, the immersion treatment with a high concentration of $ZnCl_2$-HCl solution (250 g/L) during the biochar modification process should be also responsible for the failure

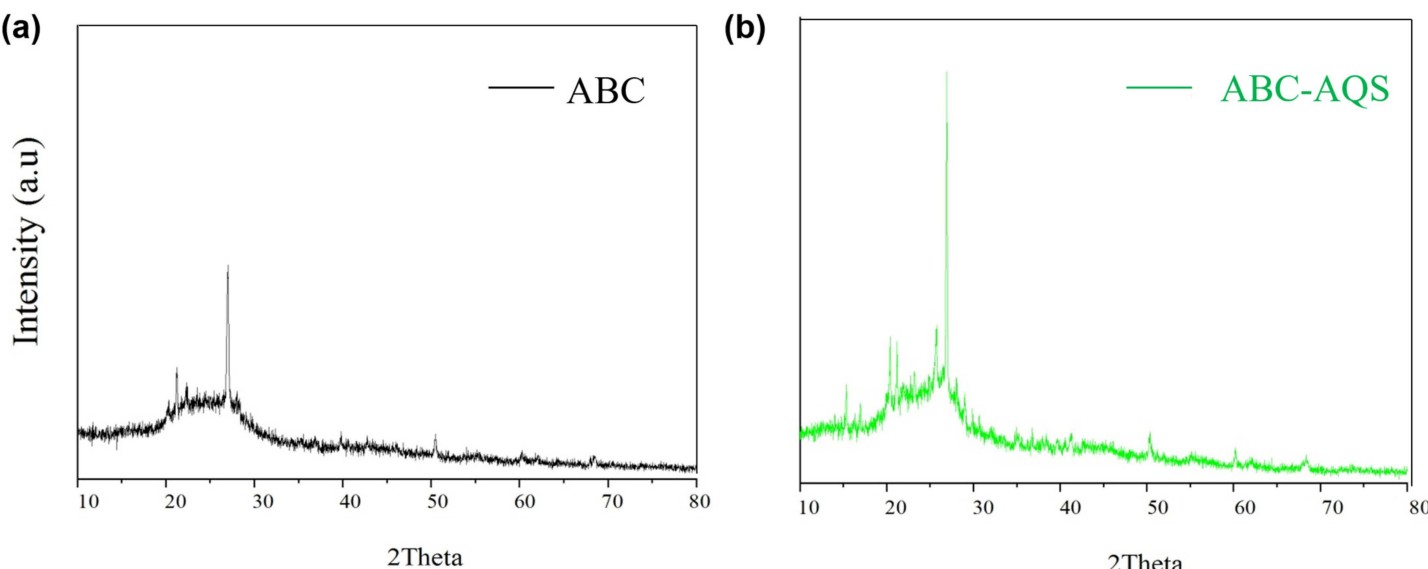

**Fig 5. The XRD patterns of the raw and the functionalized biochar. a, raw biochar; b, functionalized biochar.**

of anaerobic methanogenesis ([Fig 1a]). Although the $Zn^{2+}$ concentration in the digestate was not determined in this study, the toxic inference by $Zn^{2+}$ that originated from the AQS-functionalized biochar could be listed as one of the reasons for the methanogenic inhibition in this study. Considering the fact that $Zn^{2+}$ could be bound to the modified biochar by surface complexation and electrostatic adsorption, and eventually released into digestate during the anaerobic digestion process [34,35], the excessive $Zn^{2+}$ that originated from the biochar modification should also have posed a toxic effect on the anaerobic microorganisms during the anaerobic digestion process. Therefore, compared to the improvement of redox activity after biochar modification, the microbial inhibition caused by side reactions during chemical modification may have a stronger impact on anaerobic digestion process.

## Conclusions

Although the redox activity of algal biochar, especially the electron-donating capacity was enhanced by the immobilization of anthraquinone-2-sulfonic acid on its surface, the anaerobic methanogenesis was completely inhibited by the quinone-functionalized biochar. The possible reasons originated from the strong acid treatment involved in the modification process. Both the biochar structures and surface groups were altered due to the acid treatment process, which have posed a negative impact on the pH stability of anaerobic digestion. Moreover, the acid pretreatment process might have promoted the generation and/or dissolution of inhibitory substances (e.g., heterocyclic aromatic hydrocarbons) by the side reactions, and the release of heavy metals (e.g., $Zn^{2+}$) during anaerobic digestion with ABC-AQS, which eventually contributed to the methanogenic inhibition during anaerobic digestion. The findings of this study give us an important clue that when designing a biochar modification procedure for a given biotechnological process, the possible influences of chemical side reactions during biochar modification process on subsequent microbial metabolism should be emphasized.

## Supporting information

**S1 File. The main characteristics of different biochar materials used in this study.**
(DOCX)

## Acknowledgments

The preparation of algal biochar and anaerobic digestion test were performed at Jiangsu key laboratory of anaerobic biotechnology, Jiangnan University. Thanks for the assistance of He Liu and Qihao Cao.

## Author contributions

**Conceptualization:** Qian Jiang.

**Data curation:** Qian Jiang, Wentao Zhou, Yue Chen, Chengcheng Li.

**Formal analysis:** Zhenglong Peng, Chengcheng Li.

**Funding acquisition:** Qian Jiang.

**Investigation:** Qian Jiang, Wentao Zhou, Yue Chen, Zhenglong Peng.

**Methodology:** Qian Jiang, Zhenglong Peng, Chengcheng Li.

**Project administration:** Qian Jiang.

**Software:** Wentao Zhou, Yue Chen, Chengcheng Li.

**Supervision:** Qian Jiang.

**Visualization:** Chengcheng Li.

**Writing – original draft:** Qian Jiang.

**Writing – review & editing:** Qian Jiang.

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
