## [Decision Letter · Decision Letter 0]

12 Feb 2025

PONE-D-25-01514Impacts of the quinone-functionalized biochar on anaerobic digestion: Beyond the redox property of biocharPLOS ONE

Dear Dr. Jiang,

Thank you for submitting your manuscript to PLOS ONE. After careful consideration, we feel that it has merit but does not fully meet PLOS ONE’s publication criteria as it currently stands. Therefore, we invite you to submit a revised version of the manuscript that addresses the points raised during the review process.

We look forward to receiving your revised manuscript.

Kind regards,

Ivan P. Kozyatnyk, Ph.D.

Academic Editor

PLOS ONE

Journal Requirements:

“Suqian Sci&Tech Program (Grant NO. K202333), Scientific Research Foundation of Suqian University (NO.106-CK00042/060)”

“The preparation of algal biochar and anaerobic digestion test were performed at Jiangsu key laboratory of anaerobic biotechnology, Jiangnan University. This study was financially supported by Suqian Sci&Tech Program (Grant NO. K202333), Scientific Research Foundation of Suqian University (NO.106-CK00042/060). “

“Suqian Sci&Tech Program (Grant NO. K202333), Scientific Research Foundation of Suqian University (NO.106-CK00042/060)”

5. We note that Figure 1 in your submission contain copyrighted images. All PLOS content is published under the Creative Commons Attribution License (CC BY 4.0), which means that the manuscript, images, and Supporting Information files will be freely available online, and any third party is permitted to access, download, copy, distribute, and use these materials in any way, even commercially, with proper attribution. For more information, see our copyright guidelines: http://journals.plos.org/plosone/s/licenses-and-copyright.

Reviewers' comments:

Reviewer's Responses to Questions

**Comments to the Author**

1. Is the manuscript technically sound, and do the data support the conclusions?

Reviewer #1: Partly

Reviewer #2: Partly

2. Has the statistical analysis been performed appropriately and rigorously? 

Reviewer #1: No

Reviewer #2: Yes

3. Have the authors made all data underlying the findings in their manuscript fully available?

Reviewer #1: No

Reviewer #2: Yes

4. Is the manuscript presented in an intelligible fashion and written in standard English?

Reviewer #1: Yes

Reviewer #2: Yes

5. Review Comments to the Author

Reviewer #1: The manuscript demonstrates a highly relevant issue related to attempts of intensification of biogas production using engineered biochar. It was hypothesized that quinone functional groups could intensify methane production due to increase of redox potential of biochar. Results showed opposite influence on the overall methane production meaning its inhibition, but acidogenesis rate was improved. Previous results were presented in the Introduction section including the description of knowledge gaps. Detailed protocols and methods for chemical analysis and measurements are presented as supporting information online. Details of the methodology are sufficient to allow the experiments to be reproduced. Units of measurement were clearly defined in all tables and figures. P-values were reported for parameters of electron capacity of different types of biochar (p < 0.01). Displaying data in plots is sufficient.

At the same time, I came up with some major and minor concerns.

1. Section Introduction is well-designed for reader without specific knowledge in this area, but statement of hypothesis is lack.

2. Authors claimed accepting of PLOS’s Data Availability policy, but I could find dataset of experiment’s results in SI, and original data are not deposited in appropriate repositories.

3. Section Materials and Methods was very well described, but a paragraph about Statistical analysis should be developed. It is not clear from the text which measures of variance (standard deviation, standard error of the mean, confidence intervals) and central tendency (mean, median) were being presented.

4. Presented results related to biochar properties and methane production are relevant and important, but no regression analyses or relationships between those parameters were found.

5. There is a lack of argumentation and discussion about the choice of substrate for biochar production and its ecological safety due to waste origin and probability of containing some pollutants (line76).

6. The number of samples for analysis was not found in the manuscript.

7. It was stated that glucose served as the sole carbon source for the anaerobic digestion test to avoid possible interference from substrates (line 110), but it’s not clear if the same results of methane production will be obtained in the case of real-stated biogas reactor with common substrate.

8. In the Model fitting analysis paragraph reference to SI was stated as well as series of other reports. Which reports have been meant?

9. Sections Results and Discussion are combined, but it will be easier to read manuscript if they are separated.

10. Obtained results are explained and discussed, but practical implications are lacking as well as further research due to the rejecting of initial assumptions about the influence of modified biochar on methane production.

11. The manuscript is well organized but written not enough clearly to be accessible to non-specialists.

However, the study itself shows sufficient potential that the authors are encouraged to resubmit a revised version according to mentioned comments.

Reviewer #2: In this manuscript, anthraquinone-2-sulfonate (AQS) was immobilized on algal biochar (ABC)

by surface doping method, and the impacts of the quinone-functionalization process on algal biochar for regulating methane production were investigated. Furthermore, the authors have explored the mechanisms associated with the methanogenic performances with the addition of quinone-functionalized biochar (ABC-AQS). Interestingly, the study's findings will help to clarify the purpose and importance of the quinone-functionalization process on algal biochar in controlling methane generation, which will be helpful in the development of designed biochar for real-world uses. Overall, this is a meticulous and informative work, which is a topic of interest to the researchers in the related areas. Although the work is well designed and experiments are well organized, however, some major concerns have arisen on reading the manuscript which should be addressed.

1. The abstract is not well-organized. Regarding the abstract, it should be noted that a good abstract should contain the following points:

• The state of the field and/or the gap your research is filling

• Describing what you did to develop your argument

• The summary of major findings

• The larger implications of your findings

In the abstract of the current manuscript, the authors have mainly described what they have done and the summary of their major findings. However, the other three points are missing in the current abstract. In this aspect, the abstract is not compelling. The authors are advised to organize the abstract by discussing the above-mentioned points one by one and highlight the significance/novelty of the work. Additionally, for further improvement of the Abstract reach, I suggest that you incorporate a note in your end statement, commenting on the novelty of the work.

2. In the introduction part, please include a schematic diagram in order to get an in-depth knowledge on the purpose of the current study. It will further strengthen the novelty of the current study.

3. Statistical analysis is missing in Figure 2(b) and (c). The authors are strongly recommended to include error bars in the mentioned Figures.

4. Materials and Methods: Some experimental details are missing in the current manuscript. Various techniques described as cyclic voltammetry (CV) method and X-ray diffraction measurements should be described in detail with different parameters used for the analysis. What is the concentration used for those analysis? What is the solvent used for these studies? Please mention all these things in the methods.

5. For completeness purposes, please include a comparison of the results reported in this work and others.

6. The authors are advised to briefly address the future perspectives, limitations, and drawbacks of the current study in the conclusion part. It is not necessary to go into great depth when describing the scope of future study; a rough sketch of the areas needing additional research and/or optimization should do.

7. Please provide a descriptive supporting information section in the manuscript with at least 2-3 sentences.

6. PLOS authors have the option to publish the peer review history of their article (what does this mean? ). If published, this will include your full peer review and any attached files.

**Do you want your identity to be public for this peer review?** For information about this choice, including consent withdrawal, please see our Privacy Policy .

Reviewer #1: No

Reviewer #2: No

---

## [Author Response · Author response to Decision Letter 1]

2 Mar 2025

Response to Reviewers

Manuscript Number: PONE-D-25-01514

Manuscript Title: Impacts of the quinone-functionalized biochar on anaerobic digestion: Beyond the redox property of biochar

Article Type: Research Article

Author(s): Qian Jiang, Wentao Zhou, Yue Chen, Zhenglong Peng, Chengcheng Li

Journal Requirements:

Response: Thanks for the comments. The manuscript has been carefully revised according to the PLOS ONE style templates:

The tittle, author, affiliations formatting of the manuscript have been revised, please see Page 1, Lines 1-11; the same revisions have also been made in the “Supporting Information” file.

The manuscript body formatting has been revised, please see Page 3, Lines 35-36, 39, 42-45; Page 4, Line 59, 68; Page 5, Line 88; Page 6, Lines 106-108; Page 11, Lines 202-203; Page 12, Line 238; Page 13, Lines 240-241, 253-255, 259; Page 14, Line 275; Page 15, Lines 288-289; Page 16, Line 307; the file naming has been revised according to the guidelines.

Response: Thanks for the comments. No permits were required in this study and explanations were listed as follows: the blue algae biomass was collected by the filtration method, which has been performed by the authors of this study; the preparation of biochar was performed in a lab-scale tube furnace (Shengli Instruments, China), the modification of biochar, the batch anaerobic digestion tests, and the chemical analysis were all performed in the laboratory. Accordingly, the additional information about the “Materials and methods” was revised, please see Page 5, Lines 82-87.

“Suqian Sci&Tech Program (Grant NO. K202333), Scientific Research Foundation of Suqian University (NO.106-CK00042/060)”

Response: Thanks for the comments. The financial disclosure was stated in the cover letter: “The funders had no role in study design, data collection and analysis, decision to publish, or preparation of the manuscript.” Please see the cover letter.

“The preparation of algal biochar and anaerobic digestion test were performed at Jiangsu key laboratory of anaerobic biotechnology, Jiangnan University. This study was financially supported by Suqian Sci&Tech Program (Grant NO. K202333), Scientific Research Foundation of Suqian University (NO.106-CK00042/060). “

“Suqian Sci&Tech Program (Grant NO. K202333), Scientific Research Foundation of Suqian University (NO.106-CK00042/060)”

Response: Thanks for the comments. The funding-related text has been removed from the Acknowledgments section of manuscript. The Acknowledgments section has been revised to “The preparation of algal biochar and anaerobic digestion test were performed at Jiangsu key laboratory of anaerobic biotechnology, Jiangnan University. Thanks for the assistance of He Liu and Qihao Cao.” Please see Page 17, lines 337-338.

Funding Statement has been revised to “Funding: This research is financially supported by the Suqian Sci&Tech Program (Grant NO. K202333) and the Scientific Research Foundation of Suqian University (NO.106-CK00042/060). The funders had no role in study design, data collection and analysis, decision to publish, or preparation of the manuscript.” Please see the cover letter.

5. We note that Figure 1 in your submission contain copyrighted images. All PLOS content is published under the Creative Commons Attribution License (CC BY 4.0), which means that the manuscript, images, and Supporting Information files will be freely available online, and any third party is permitted to access, download, copy, distribute, and use these materials in any way, even commercially, with proper attribution. For more information, see our copyright guidelines: http://journals.plos.org/plosone/s/licenses-and-copyright.

Response: Thanks for the comments. Figure 1 has been removed from the manuscript, and the detailed information for the preparation of algal biochar in the removed figure has been added and revised in the manuscript, please see Page 5, Lines 82-90.

For the rest of figures, captions and citations were all revised accordingly. Please see Page 5, line 91; Page 6, lines 98, 106-108; Page 9, line 175; Page 10, Lines 177, 179, 185, 192-194; Page 11, lines 200-203, 207, 210, 213; Page 12, lines 220, 223-225, 233, 235; Page 13, lines 240-241, 244, 246, 247, 253-255; Page 14, line 277; Page 15, lines 288-289, 292, 294, 314.

Reviewer’s comments:

Reviewer #1:

1. General comments

Reviewer #1: The manuscript demonstrates a highly relevant issue related to attempts of intensification of biogas production using engineered biochar. It was hypothesized that quinone functional groups could intensify methane production due to increase of redox potential of biochar. Results showed opposite influence on the overall methane production meaning its inhibition, but acidogenesis rate was improved. Previous results were presented in the Introduction section including the description of knowledge gaps. Detailed protocols and methods for chemical analysis and measurements are presented as supporting information online. Details of the methodology are sufficient to allow the experiments to be reproduced. Units of measurement were clearly defined in all tables and figures. P-values were reported for parameters of electron capacity of different types of biochar (p < 0.01). Displaying data in plots is sufficient.

At the same time, I came up with some major and minor concerns.

Response: Thanks for the comments. The manuscript has been carefully revised according to the comments. The details of the revisions have been listed below following each specific comment, point-by-point. The response to each comment is in blue in this file; all of the errors as we know have been corrected and marked with the red color in the revised manuscript.

2. Special comments

1. Section Introduction is well-designed for reader without specific knowledge in this area, but statement of hypothesis is lack.

Response: Thanks for the comment. The section Introduction of this manuscript has been revised, and the statement of hypothesis has been added. Please see Page 4, Lines 63-70.

2. Authors claimed accepting of PLOS’s Data Availability policy, but I could find dataset of experiment’s results in SI, and original data are not deposited in appropriate repositories.

Response: Thanks for the comment. The authors confirm that all data underlying the findings are within the paper and its Supporting Information files. In the data statement we have described where the data may be found in full sentences: All relevant data are within the manuscript and its Supporting Information files. Thanks again for your suggestion.

3. Section Materials and Methods was very well described, but a paragraph about Statistical analysis should be developed. It is not clear from the text which measures of variance (standard deviation, standard error of the mean, confidence intervals) and central tendency (mean, median) were being presented.

Response: Thanks for the comment. The description of the statistical analysis of data in this study has been revised, please see in the section Materials and Methods, Page 9, Lines 166-171.

4. Presented results related to biochar properties and methane production are relevant and important, but no regression analyses or relationships between those parameters were found.

Response: Thanks for the comment. Yes, we all agree that biochar properties and methane production are relevant and important. However, results in this study have showed opposite influence on the overall methane production by the modified biochar, therefore the regression analyses or relationships between biochar properties and methanogenic improvement were not performed. Instead the main purpose of this study was to clarify the role and significance of the quinone-functionalization process on algal biochar in controlling methane generation, the study's findings will help to the development of designed biochar for real-world uses. To make it clear, the implications of this work has been added and revised, pleased see Page 17, Lines 332-334.

5. There is a lack of argumentation and discussion about the choice of substrate for biochar production and its ecological safety due to waste origin and probability of containing some pollutants (line76).

Response: Thanks for the comment. Firstly, Taihu blue algae, a kind of abundant and harmful biomass waste in the water treatment of Wuxi, China. Till 2018, the yearly refloated Taihu blue algae reached about 1.87 million tons, while the refloated Taihu blue algae of Wuxi city since 2007 totaled about 13.43 million tons, consisting 90% of the algal amounts refloated in Jiangsu Province, China. It is of great important significance to develop an approach for the treatment and disposal of the Taihu blue algae biomass;

Secondly, the preparation of biochar from the Taihu blue algae has been consistently studied by our team, a series of works has been published in the nearly 5 years, eg., 10.1016/j.biortech.2021.125493; 10.1016/j.enconman.2022.115417; 10.1016/j.jece.2023.109850; 10.1016/j.jwpe.2024.106123. Moreover, algal biomass originated biochar were found to possess significantly higher N-content than that from the lignocellulosic biomass, which would in turn promote the electrochemical traits for the electron transfer during the biochemical process. Therefore, the Taihu blue algae was continued to serve as the precursor for biochar production in this study, hoping to further expand the treatment and disposal of this solid waste;

Lastly, according to our previous study, as well as other study (Shi et al., 2015, Environmental science & Technology), the dominated community of the Taihu blue algal biomass obtained was Microcystis; moreover, algal biomass was collected from the algal blooming sites alongside Taihu Lake of Wuxi city, filtration and cold drying method were employed to avoid chemicals addition during the biomass harvest; the obtained algae powder were stored at 4 oC before use. Therefore, we confirm that the probability of some pollutants being present in algal biomass should be low in this study.

Based on the above consideration, the description the choice of substrate for biochar production in this manuscript has been added and revised, please see Page 5, Lines 82-90.

6. The number of samples for analysis was not found in the manuscript.

Response: Thanks for the comment. The description of samples for analysis in this study could be found in the section Materials and methods and the section Results and discussion:

(1) The anaerobic digestion tests were conducted in eight customized glass reactors (1.2 L) with working volume of 600 mL, both the gas and liquid samples were analyzed during the batch experiments. After first batch of anaerobic digestion, the second batch of anaerobic digestion was repeated under the same condition as the first batch. The resulted data obtained from the above two batch experiments were analyzed and plotted in Figure 2 of this study. Please see Page 8, Lines 145-148;

(2) For the analysis of biochar samples, inclduing pH, CV, and electron exchange capacity (EEC), the number of samples were set to a minimum of 3 and the resultant data was showed in the Figures and the S1_Table. For example, in Figure 4, the number of samples for quantitative EEC analysis was 6, the raw data point and the statistical analysis results were displayed directly in the image.

7. It was stated that glucose served as the sole carbon source for the anaerobic digestion test to avoid possible interference from substrates (line 110), but it’s not clear if the same results of methane production will be obtained in the case of real-stated biogas reactor with common substrate.

Response: Thanks for the comment. Yes, we all agree that glucose served as the sole carbon source for the anaerobic digestion test to avoid possible interference from substrates. This is because the main purpose of this study was to explore the impact of modified biochar, for which the effect of other factors in the anaerobic reactors needed to be minimized as much as possible. This strategy has also been adopted by other researchers, eg., http://doi.10.1021/acs.est.0c00112.

Moreover, we are also very interested in the impacts of quinone-functionalized biochar on the real-stated biogas reactor with common substrate. The mechanisms and feasibility of the addition of functionalized biochar to improve the anaerobic digestion performance are still worthy o

---

## [Decision Letter · Decision Letter 1]

19 Mar 2025

Impacts of the quinone-functionalized biochar on anaerobic digestion: Beyond the redox property of biochar

PONE-D-25-01514R1

Dear Dr. Jiang,

We’re pleased to inform you that your manuscript has been judged scientifically suitable for publication and will be formally accepted for publication once it meets all outstanding technical requirements.

Kind regards,

Ivan P. Kozyatnyk, Ph.D.

Academic Editor

PLOS ONE

Additional Editor Comments (optional):

Reviewers' comments:

Reviewer's Responses to Questions

**Comments to the Author**

1. If the authors have adequately addressed your comments raised in a previous round of review and you feel that this manuscript is now acceptable for publication, you may indicate that here to bypass the “Comments to the Author” section, enter your conflict of interest statement in the “Confidential to Editor” section, and submit your "Accept" recommendation.

Reviewer #1: All comments have been addressed

Reviewer #2: All comments have been addressed

2. Is the manuscript technically sound, and do the data support the conclusions?

Reviewer #1: Yes

Reviewer #2: Yes

3. Has the statistical analysis been performed appropriately and rigorously? 

Reviewer #1: Yes

Reviewer #2: Yes

4. Have the authors made all data underlying the findings in their manuscript fully available?

Reviewer #1: Yes

Reviewer #2: Yes

5. Is the manuscript presented in an intelligible fashion and written in standard English?

Reviewer #1: Yes

Reviewer #2: Yes

6. Review Comments to the Author

Reviewer #1: Dear authors,

thank you for your detailed responses on each of my questions and comments.

I satisfy with answers.

Reviewer #2: The authors have addressed all the comments and revised the manuscript as suggested. Therefore, it is recommended for publication.

7. PLOS authors have the option to publish the peer review history of their article (what does this mean? ). If published, this will include your full peer review and any attached files.

**Do you want your identity to be public for this peer review?** For information about this choice, including consent withdrawal, please see our Privacy Policy .

Reviewer #1: No

Reviewer #2: **Yes: ** Dr. Pooja Ghosh

---

## [Editor Report · Acceptance letter]

PONE-D-25-01514R1

PLOS ONE

Dear Dr. Jiang,

I'm pleased to inform you that your manuscript has been deemed suitable for publication in PLOS ONE. Congratulations! Your manuscript is now being handed over to our production team.

Kind regards,

on behalf of

Dr. Ivan P. Kozyatnyk

Academic Editor

PLOS ONE
